# Exploring the Impact of Extended Maceration on the Volatile Compounds and Sensory Profile of Monastrell Red Wine

Alejandro Martínez-Moreno [1],* , Rosa Toledo-Gil [2] , Ana Belén Bautista-Ortin [1] , Encarna Gómez-Plaza [1] , José Enrique Yuste [2] and Fernando Vallejo [2]

[1] Department of Food Science and Technology, Faculty of Veterinary Sciences, University of Murcia, Campus de Espinardo, 30100 Murcia, Spain; anabel@um.es (A.B.B.-O.); encarna.gomez@um.es (E.G.-P.)

[2] Metabolomics Platform, Centro de Edafología y Biología Aplicada del Segura (CEBAS-CSIC), Campus Universitario de Espinardo, 30100 Murcia, Spain; rtoledo@cebas.csic.es (R.T.-G.); jyuste@cebas.csic.es (J.E.Y.); fvallejo@cebas.csic.es (F.V.)

* Correspondence: martinezamoreno@gmail.com; Tel.: +34-868887323

**Abstract:** Volatile organic compounds (VOCs) are crucial to the wine's overall quality since they define the aromatic profile. The aim of this study was to investigate whether a 146-day extended maceration (EM) treatment positively affects the aromatic and sensory properties of Monastrell red wine. A total of 43 aromatic compounds belonging to different chemical classes were identified using solid-phase microextraction combined with gas chromatography–mass spectrometry (SPME/GC-MS). In general, EM treatment decreased both the number and total relative concentration of VOCs. Specifically, EM decreased the concentration of alcohols, terpenes and sulphur compounds compared to control wines. However certain compounds such as 2-ethyl-1-hexanol, phenylethyl and ethyl decanoate significantly increased with prolonged maceration. Conversely, EM treatment did not significantly affect the total relative concentrations of esters and ketones. From sensorial point of view, the triangular test showed a positive identification of wines (10/18) with a significant preference for EM wines. Moreover, descriptive analysis revealed that EM wines scored lower values in appearance, aroma and taste. Future research should aim to optimize maceration time to enhance the content of VOCs without compromising the sensory quality of the wine.

**Keywords:** VOCs; SPME/GC-MS; descriptive analysis; maceration time

## 1. Introduction

Monastrell, also known as Mourvèdre, is an autochthonous red grape variety primarily grown in the southeast of Spain, particularly in the Murcia region. It thrives in hot, dry climates and produces robust, full-bodied wines with high tannins and deep colour being a key variety in both Spanish and Southern French wines, valued for its structure and aging potential [1]. Wine aroma is one of the most important characteristics of red wines, significantly contributing to their quality. It is produced by a complex balance of different chemical classes of volatile organic compounds (VOCs), belonging to alcohols, esters, aldehydes, lactones, terpenes, $C_{13}$-norisoprenoids, volatile phenols, fatty acids, carbonyls, sulphur and nitrogen compounds [2]. All these VOCs present in wines are derived from fruit, through the fermentation and aging processes [3] contributing to the overall aroma of red wine [4]. Thus, Monastrell is known for its rich flavours of dark fruit, herbs, and spice, often with earthy undertones. However, VOC fingerprint is determined by a complex number of variables, such as grape variety (grape-derived volatiles play a crucial role in determining the varietal flavour), degree of maturation, climatic conditions, cultivation practices and finally winemaking processes like maceration, fermentation temperature, etc. [2].

In this line, maceration is the stage of red wine production in which grape solids, including seeds and skins, remain in contact with the must/wine [5]. Thus, it is probably

the most important phase to produce high quality red wines since their most salient sensory features, like the phenolic and aromatic profile, will be defined [5]. Typically, maceration time is limited to the duration of alcoholic fermentation (7–15 days). Indeed, winemakers traditionally use the end of alcoholic fermentation as the cut-off point for how long solids should remain in contact with the wine [5]. In traditional winemaking, different maceration techniques include (i) conventional maceration consisting of moving the must from the bottom of the vat to the top or immersing the floating layer of skins; (ii) pre-fermentation maceration or "cold soaking", in which grape skins and seeds are put in contact with the liquid must at low temperatures one or two days before fermentation begins; (iii) carbonic maceration, which is carried out with whole grapes fermented in a $CO_2$-rich environment; and (iv) post-fermentation or extended maceration, which involves prolonging the contact of grape solids with the wine after wine fermentation is completed [6].

In recent years, extended maceration (EM) has gained popularity worldwide. The duration of EM is varying widely, ranging from few days [7] to some months [5]. Several studies have investigated the impact of EM on the composition and quality of both red and white wines, focusing predominantly on basic wine components and chromatic and phenolic attributes [5–9].

Previous studies have shown a large variability in the effects of EM on red wine composition. The "optimal" maceration length depends basically on the variety and the season, with the grape phenolic composition and skin extractability being key factors [8]. Generally, EM decreased the chromatic, anthocyanin and polymeric pigment composition of the resulting wines [5]. It increases the concentration of seed-derived tannins, especially catechins and proanthocyanidins in Merlot [10] and Cabernet Sauvignon [11] and the total phenols, leading to higher antioxidant capacity [12,13]. From a sensory standpoint, different studies have reported that EM increases bitterness and astringency in the finished wines [11,13]. Thus, in an EM study of the white wines Minutolo and Verdeca [14], a different volatile profile was observed in the macerated wines, including some oxidation and aging compounds that were not present in the control wines. Furthermore, maceration successfully enhanced varietal identity of wines, improving terpenes composition. In a 30-day EM experiment, ref. [2] reported significant changes in the aroma profile of Vranec wines due to maceration time. The authors found that EM increased the volatile compounds up to the seventh day and then a decreasing was observed after 30 days of maceration. Another 90-day EM study on Aglianico di Taurasi wines showed a significant increase in VOC families (alcohols, esters, carbonyl compounds, fatty acids, lactones and phenols) as EM increased [6]. Conversely, ref. [9] reported a decrease in different VOCs such as esters, terpenes and others (vitispirane and b-damascenone). However, these authors observed a significant increase in volatile alcohol concentration after 8 weeks of EM in Merlot wines. Other studies conducted on red wines have also reported a significant reduction in the concentration of most identified volatile compounds as the duration of extended maceration increases [7].

All the above studies described high variability in VOC fingerprint. Therefore, understanding how different EM lengths affect each variety is crucial for determining the optimal maceration duration to extract aroma compounds. As far as we know, no information is available on the effect of EM on Monastrell red wine VOCs. In this context, the main goal of this study was to evaluate the influence of long extended maceration on the presence and concentration of different VOCs in Monastrell red wines. Additionally, the organoleptic quality was assessed to optimize maceration time for producing better wines.

## 2. Materials and Methods

### 2.1. Grape Samples and Winemaking

First, 300 kg of Monastrell grapes were manually harvested and transported to the University of Murcia experimental winery. The average grape berry was 1.92 g and the fruit composition at harvest were as follows: total soluble solids was 24.6 °Brix, pH was 3.14 and tartaric titratable acidity was 5.25 $gL^{-1}$. Grapes were maintained for one day in a

cold room ($4 \pm 2\ °C$). All grapes were destemmed, crushed and distributed into six 50 L tanks (2 treatments $\times$ 3 replicates each one). Sulphur dioxide ($SO_2$) was added at a rate of 30 mg/kg during the fermenter filling process. Musts were inoculated with selected dry yeast (Viniferm CT007, Agrovin, Spain) at a rate of 20 $gh^{-1}\ L^{-1}$. All vinifications were conducted at $24 \pm 2\ °C$ and sugar consumption was monitored daily with a densitometer (DMA 35N, Anton Paar, Graz, Austria). Cap management consisted of a punch down twice a day during active fermentation (10 days). After the end of alcoholic fermentation, control wines were pressed and stored again in the same tanks. All wines remained in these tanks at a controlled temperature of $21\ °C \pm 2$ for the allotted duration of the experiment (146 days). After the end of alcoholic fermentation, all tanks were sulphited (35 $mgL^{-1}$) and dry ice was added every two weeks in all tanks to minimize contact with oxygen and the risk of microbial spoilage during the extended maceration treatment. The physiochemical parameters of Monastrell wines after fermentation were 14.8% alcohol by vol, 5.0 $gL^{-1}$ total acidity, pH of 3.31 and 1.23 $gL^{-1}$ of malic acid. Malolactic fermentation was not performed during extended maceration time and after maceration finished, EM wines were pressed and stored in the same tanks. After one week, all wines were filtered and sulphited (30 $mgL^{-1}$) and bottled in Bordeaux bottles with high-quality brown glass and cork closures.

### 2.2. SPME-GC–MS Analyses

An automated solid-phase microextraction (SPME) combined with gas chromatography–mass spectrometry (GC–MS) is a highly efficient separation technique for extraction and separation of wine aroma compounds, in accordance with the method described by [15]. Ten milliliters of each sample was taken from each wine and added to 20 mL vials, and then 25 µL of an internal standard 2-octanol (250 ppb $L^{-1}$) (Sigma-Aldrich, Madrid, Spain) and 3 g of NaCl were added to each one before loading onto an auto-sampling tray (Gerstel MPS Dual Head RobPro/Rob 160 cm. KG, Mellinghofen, Germany). All samples were analysed in triplicate.

Gas chromatographic analyses were carried out on an HP 8890B gas chromatography (GC) system coupled to an HP 5977B quadrupole mass spectrometer (Agilent Technologies, Palo Alto, CA, USA). Prior to the extraction of the volatiles, the samples were equilibrated at room temperature at $25\ °C$ for 30 min. For the SPME, a DVB/Carboxen/PDMS 50/30, stable flex-gray fibre was used (Supelco, Bellefonte, PA, USA). The first step was to bakeout it for 3 min at $250\ °C$. After that, it was penetrated 28 mm into the headspace of the 20 mL vial for 10 min. Immediately after the exposure, the fibre was transferred to the GC-injector for 5 min of thermo-desorption at $270\ °C$ with an inlet mode of split (ratio 1:10) through an ultra-inert liner (5190-2292-900 µL, Agilent Technologies). An analytical column of medium polarity (VF-WAXms, 30 m $\times$ 0.25 mm $\times$ 0.25 µm, Agilent Technologies) was used with the following temperature program: $40\ °C$ for 5 min with a temperature ramp of $4\ °C\ min^{-1}$ up to $250\ °C$ (holding time 1 min). The mass selective detection was performed in the scan mode (30–500 amu, EI (70 eV), detector temperature $230\ °C$, frequency of 1,6 scans/s and a cycle time of 622.9 ms).

Data were processed using the Mass Hunter Qualitative Analysis software (version B.10.00, Service Pack 1, Agilent Technologies Palo Alto, CA, USA). To find compounds, the algorithm "compound discovery by deconvolution" was carried out, selecting peaks of both GC and MS spectra with an absolute height greater than or equal to 5000 counts. Peak tentative identification was carried out by comparing mass spectra with those of the mass library (Wiley11N17 main/NIST20 mass spectral library; Wiley, Chichester, UK) with a score greater than 80% and comparing the calculated retention indices with those published in the literature. Semi-quantitative data were obtained by calculating the relative peak area (total ion count signal or that of selected fragments in the case of some coeluted compounds) in relation to the internal standard.

### 2.3. Sensory Analysis

Monastrell EM red wines were subjected to a triangular and descriptive sensory test (colour, aroma and flavour) after twelve months of aging in the bottle. Prior to the sensory analysis, bottles from the different replications were pooled to have a representative sample of the treatment. Eighteen regular wine consumers interested in the project were selected for the sensory analysis. To carry out the triangular analysis, the method described in [16] was followed. Regarding descriptive analysis, the intensity of wine colour, flavour-by-mouth attributes (mouth intensity, quality, body, persistence, bitterness and astringency) and aroma attributes (aroma intensity, aroma quality, fruit, vegetal) were scored on a scale of 0 to 10. On this scale, 0 indicates that the attribute has not been perceived by the consumers, while 10 means that the attribute is perceived very intensely.

### 2.4. Statistical Analysis

Data are expressed as the means $\pm$ standard deviations of quintupled tests. Unpaired T-tests were carried out on SPSS 26.0 (SPSS Inc., Chicago, IL, USA) to compare the significant difference among the mean at a $p \leq 0.05$ level.

### 3. Results and Discussion

After twelve months in bottles, the aroma profile of Monastrell wines after 146 days of maceration was determined by SPME/GC-MS. In this study, a total of 43 individual volatile organic compounds (VOCs) were identified quantitatively and qualitatively (Table 1), including alcohols, esters, organic acids, ketones, terpenes as well as volatile sulphur compounds.

**Table 1.** Mean values and relative standard deviation of volatile compounds ($\mu$g of 2-octanol equivalents$\cdot$L$^{-1}$) measured in wines from control and extended maceration.

| Volatile Compounds | RI [1] | RT (min) [2] | Treatments (mg$\cdot$L$^{-1}$) Control | Extended Maceration | Score [3] |
|---|---|---|---|---|---|
| **Alcohols** | | | | | |
| 2-Methyl-1-propanol | 1191 | 8.6 | 836.2 $\pm$ 53.6 a | 325.5 $\pm$ 33.0 b | 88.4 |
| 3-Methyl-1-butanol acetate | 1202 | 9.5 | 1872.7 $\pm$ 58.4 a | - | 97.1 |
| 2-Ethyl-1-butanol | 1402 | 16.5 | 13.4 $\pm$ 0.6 a | - | 82.8 |
| 4-Methyl-1-pentanol | 1447 | 16.7 | 18.0 $\pm$ 2.5 a | - | 83.6 |
| (S)-(+)-3-Methyl-1-pentanol | 1457 | 17.1 | 26.0 $\pm$ 6.5 a | 31.6 $\pm$ 2.2 a | 87.6 |
| 1-Hexanol | 1463 | 18.1 | 715.9 $\pm$ 31.6 a | 603.4 $\pm$ 53.4 b | 94.1 |
| 4-Methyl-1-hexanol | 1514 | 21.5 | 16.8 $\pm$ 0.9 a | 18.0 $\pm$ 1.6 a | 85.8 |
| 1-Heptanol | 1565 | 21.5 | 16.6 $\pm$ 0.9 a | - | 85.4 |
| 2-Ethyl-1-hexanol | 1599 | 22.6 | 17.2 $\pm$ 1.5 b | 42.8 $\pm$ 6.0 a | 83.2 |
| (S)-3-Ethyl-4-methylpentanol | 1609 | 23.1 | 11.6 $\pm$ 1.4 a | - | 85.3 |
| 2,3-Butanediol | 1625 | 24.3 | 292.3 $\pm$ 45.8 a | 167.2 $\pm$ 20.4 b | 95.2 |
| [S-(R*,R*)]-2,3-Butanediol | 1628 | 25.5 | 140.7 $\pm$ 32.9 a | 27.4 $\pm$ 4.2 b | 91.5 |
| Phenylethyl alcohol | 2112 | 34.5 | 1642.7 $\pm$ 87.8 a | 1781.2 $\pm$ 258.1 a | 95.9 |
| Total alcohols (mg$\cdot$L$^{-1}$) | | | 5934.3 $\pm$ 402.3 a | 3322.6 $\pm$ 378.8 b | |
| **Esters** | | | | | |
| Methyl acetate | 964 | 2.6 | 23.6 $\pm$ 2.4 a | 17.4 $\pm$ 0.2 b | 88.0 |
| Ethyl orthoformate | 1014 | 3.2 | - | 51.0 $\pm$ 14.0 a | 87.0 |
| Ethyl propanoate | 1047 | 4.4 | 240.0 $\pm$ 83.6 a | 82.6 $\pm$ 12.2 b | 92.0 |

**Table 1.** *Cont.*

| Volatile Compounds | RI [1] | RT (min) [2] | Treatments (mg·L$^{-1}$) | | Score [3] |
|---|---|---|---|---|---|
| | | | Control | Extended Maceration | |
| Ethyl isobutyrate | 1071 | 4.6 | 215.0 ± 35.8 a | 263.4 ± 34.6 a | 94.6 |
| Isobutyl acetate | 1107 | 5.8 | 67.6 ± 5.2 a | 32.4 ± 2.0 b | 92.2 |
| Ethyl butanoate | 1128 | 6.5 | 242.0 ± 9.2 a | 196.4 ± 8.0 b | 96.5 |
| Ethyl 2-methylbutanoate | 1173 | 7.0 | 99.1 ± 8.0 b | 128.2 ± 9.6 a | 94.3 |
| Ethyl 3-methylbutanoate | 1175 | 7.6 | 228.4 ± 31.6 a | 242.0 ± 13.0 a | 95.3 |
| Isoamyl acetate | 1215 | 9.5 | - | 647.6 ± 40.0 a | 95.3 |
| Ethyl hexanoate | 1332 | 13.6 | 2466.4 ± 198.0 a | 1914.2 ± 128.0 b | 97.1 |
| Ethyl 2-hydroxypropionate | 1349 | 17.8 | 730.2 ± 98.8 a | 274.0 ± 8.2 b | 92.1 |
| Ethyl octanoate | 1440 | 20.6 | 1872.0 ± 157.0 a | 1278.0 ± 62.0 b | 97.5 |
| Isopentyl hexanoate | 1460 | 21.4 | - | 5.4 ± 0.2 a | 82.8 |
| Ethyl decanoate | 1610 | 26.9 | 169.0 ± 16.0 b | 201.6 ± 8.4 a | 91.1 |
| Diethyl succinate | 1690 | 28.1 | 915.4 ± 38.6 a | 1094.8 ± 174.4 a | 97.7 |
| Total esters (mg·L$^{-1}$) | | | 7268.7 ± 684.2 a | 6429.0 ± 514.8 a | |
| **Acids** | | | | | |
| 2- methylpropanoic acid (isobutyric acid) | 1581 | 25.7 | - | 15.4 ± 2.2 a | 85.3 |
| Butanoic acid | 1637 | 27.4 | 12.2 ± 1.4 a | 14.0 ± 9.6 a | 81.9 |
| 3-methylbutanoic acid (isovaleric acid) | 1680 | 28.5 | 46.2 ± 4.6 a | 38.6 ± 6.2 a | 81.5 |
| Hexanoic acid | 1849 | 28.5 | 45.0 ± 3.0 a | - | 86.3 |
| 2-methyl-2-phenylethyl ester—Propanoic acid | 1877 | 31.9 | - | 10.0 ± 0.2 a | 80.1 |
| Pentanoic acid | 1944 | 33.3 | 46.8 ± 6.0 a | 25.2 ± 6.0 b | 82.9 |
| Octanoic acid | 2050 | 38.6 | 44.6 ± 7.0 a | 16.0 ± 3.6 b | 82.7 |
| Total acids (mg·L$^{-1}$) | | | 194.8 ± 22.0 a | 119.2 ± 27.8 b | |
| **Aldehydes and Ketones** | | | | | |
| 2,3-Butanedione | 955 | 4.9 | - | 10.8 ± 6.0 a | 82.0 |
| 2-Butanone | 1115 | 8.3 | - | 4.0 ± 2.6 a | 83.7 |
| 2-Octanone | 1287 | 15.4 | 22.8 ± 4.8 a | - | 83.6 |
| Total ketones (mg·L$^{-1}$) | | | 22.8 ± 4.8 a | 14.8 ± 8.6 a | |
| **Sulphur compounds** | | | | | |
| Dimethyl sulphide | 857 | 2.2 | 11.2 ± 1.0 a | 8.4 ± 0.4 b | 90.4 |
| Furfural | 1464 | 21.7 | 133.6 ± 26.0 a | 101.8 ± 6.4 b | 95.0 |
| Total sulphur compounds (mg·L$^{-1}$) | | | 144.8 ± 27.0 a | 110.2 ± 6.8 b | |
| **Terpenes** | | | | | |
| D-Limonene | 1204 | 12.1 | 34.4 ± 2.0 a | - | 81.5 |
| Eugenol | 2167 | 40.7 | 17.4 ± 4.4 a | - | 83.6 |
| Total terpenes (mg·L$^{-1}$) | | | 51.8 ± 6.4 a | - | |
| **Others** | | | | | |
| Butyrolactone | 1626 | 26.7 | 42.8 ± 4.4 a | 34.2 ± 2.6 b | 89.6 |

Data are mean ± standard deviation of triplicate tests. Different letters in each row represent significant differences at $p \leq 0.05$. [1] Kovats index reported in the literature for equivalent capillary columns (https://webbook.nist.gov/). [2] Retention time in minutes. [3] VOCs identified by matching the obtained mass spectra with the reference compounds spectra in Wiley/NIST databases with a percentage above 80%.

### 3.1. Characterization of Volatile Signature of Monastrell Wines by Maceration Using SPME/GC–MS

Aroma is an important quality criterion for wines. Thus, the identification of VOCs responsible for their aroma is considered to be a key factor for quality and authentication control [15]. The SPME/GC-MS method was applied to establish the volatile signature of wine (control) and macerated wines. A total of 43 VOCs were identified in wine samples (Table 1) belonging to different chemical families, namely ethyl esters (15), alcohols (13), organic acids (7), ketones (3), terpenes (2), sulphur compounds (2) and volatile lactone (1). These VOCs have been already identified by matching the obtained mass spectra with the reference compounds spectra in Wiley/NIST databases with a percentage above 80%.

It was highlighted that 26 VOCs were common in all wine samples analysed, namely ethyl esters (12), alcohols (8), organic acids (4), sulphur compounds (2) and a volatile lactone (1) (Table 1). Moreover, some new VOCs not found in control wines, mainly ethyl esters (ethyl-orthoformate, isoamyl-acetate, isopentyl-hexanoate), acids (isobutyric acid and 2-methyl-2-phenylethyl ester-propanoic acid) and ketones (2,3-butanedione and 2-butanone) appear as result of the maceration process (Table 1). The distribution of VOCs, according to its chemical family, is represented in µg of 2-octanol equivalents·L$^{-1}$ (Figure 1) and percentage (Figure A1).

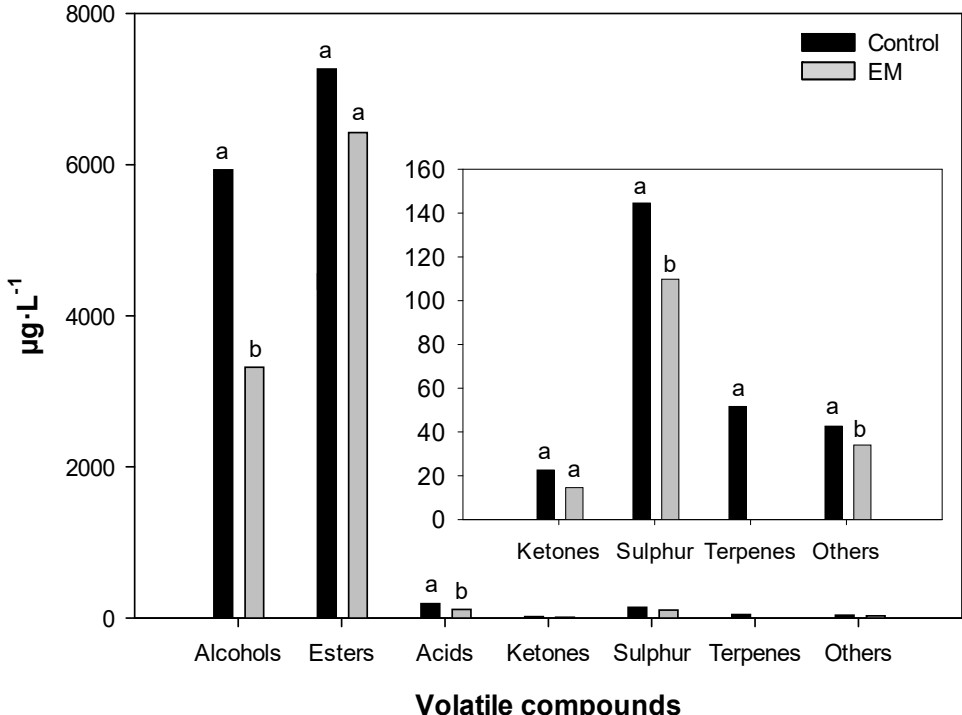

**Figure 1.** Distribution of VOCs according to their main classes identified in control and EM wines by solid-phase microextraction SPME/GC-MS expressed in µg·L$^{-1}$. Different letters in each row represent significant differences at $p \leq 0.05$.

Alcohols constitute a broad group of compounds primarily originating during alcoholic fermentation as byproducts of yeast metabolism [2]. However, some of them may also derive from grapes, either in odour-active form or as nonvolatile precursors, predominantly glycoconjugates [15]. The results revealed a higher number of alcohols identified in the control wines (13) compared to those detected in EM wines (8). Additionally, control wine exhibited a higher relative concentration of total alcohols (5934 µg L$^{-1}$). All volatile alcohols identified had a significantly higher concentration in the control wine except for 2-ethyl-1-hexanol, which was more abundant in the extended-maceration wine (Table 1). The most abundant alcohols in control wine were 3-methyl-1-butanol acetate (1872 µg L$^{-1}$) and phenylethyl alcohol (1642 µg L$^{-1}$), while in the EM wine the alcohols with the highest concentration were phenylethyl alcohol and 1-hexanol (Table 1). Phenylethyl alcohol is one

of the most important phenol-derived higher alcohols characterized by one of the most popular and desired fragrances (rosy scent) [17], and it has been described as a volatile alcohol with a floral and rosy-like odour. This compound has been notably identified previously in significant quantities within red wines as a Monastrell [18,19]. Moreover, this compound has been previously reported in others varieties as Minutolo and Verdeca [14]. Furthermore, agreeing with our results, these authors reported an increase of phenylethyl alcohol in the Minutolo wines after six months of EM. However, ref. [9] reported a general increase in volatile alcohols with the exception of phenethyl alcohol after 8 weeks of EM in Merlot wines.

The concentration of 1-hexanol was significantly higher in the control wine. This alcohol belongs to a C6-alcohols group that can be originated from grapes and usually are formed in pre-fermentative steps such as harvesting, crushing or grape maceration [20]. C6-alcohols are known for their distinct "green" aroma reminiscent of leaves and freshly cut grass [18]. They are formed from the corresponding C6-aldehydes by alcohol dehydrogenase enzymes [21]. The higher concentration of 1-hexanol in control wine might negatively impact the wine aroma, since this volatile compound is correlated with green and herbaceous odour descriptors [22] (Table 1). Another EM study on Karaoğlan wines reported a significant decrease in 1-hexanol in wines after fifteen days of EM [7]. In this sense, ref. [2] also reported slight decrease in Vranec wines after 30 days of EM.

Volatile esters are one of the most important families of aroma compounds in wines and are largely responsible for primary or fruity aromas in red wines [23]. Our results show that Monastrell wines are rich in esters, as 15 esters were identified. Furthermore, esters obtained the highest volatile compound concentration with a total of 7268 µg L$^{-1}$ in the control wine and 6429 µg L$^{-1}$ in the EM wine (Table 1). Among them, ethyl hexanoate and ethyl octanoate were dominant compounds in both wines from this fraction, followed by diethyl succinate. EM significantly increased the concentrations of ethyl 2-methylbutanoate and ethyl decanoate. Additionally, there were three esters that were only detected in the extended-maceration wines: ethyl orthoformate, isoamyl acetate and isopentyl hexanoate. EM significantly reduces the concentration of methyl acetate, ethyl propanoate, isobutyl acetate, ethyl butanoate, ethyl hezanoate and ethyl 2-hydroxypropionate.

Esters are key volatile components in wine aroma, contributing positively to the sensory properties and aromatic profile of wines. Their formation and content in wine primarily depend on the number of alcohols and acids present [2]. Their source originates from fermentation, a byproduct of yeast metabolism. This process involves esterification reactions between alcohols and fatty acids, aimed at detoxifying higher alcohols produced under stressful conditions, such as nitrogen limitation [7]. With Monastrell wines containing numerous alcohols and acids, ester formation is expected to be abundant, as seen in this study. Additionally, our findings showed that EM in Monastrell wines could influence individual ester production, modifying the aromatic profile of the wines, without forgetting that the total ester concentration was slightly lower (statistically non-significant) in EM wines. Regarding the esters detected in extended macerated wines, the literature reports conflicting results. On the one hand, the concentration of esters was found to decrease with extended time of maceration, which agrees with a number of previous reports [2,7,9,24] and disagrees with some others [6,14,25–27]. In general, our results showed a slight decrease in the concentration of esters in EM wines. This reduction in concentration observed in wines undergoing EM may be due to the prolonged contact time with lees, which have the ability to absorb esters and release esterase that hydrolyses them [24].

Within the family of acids, seven compounds were detected in Monastrell wines, including isobutyric, butanoic, isovaleric, hexanoic, propanoic, octanoic and pentanoic acids. These compounds are predominantly fermentation products, although the can also be detected in grapes [21]. Among them, isovaleric and pentanoic acids were the main ones, with concentrations of 46.2 µg L$^{-1}$ and 46.8 µg L$^{-1}$ in control wine, respectively. Isobutyric acid was not found in the control wine and hexanoic acid was not identified in the EM wine (Table 1). Out of all the identified acids, only butanoic acid was found in

higher quantities in EM wines, although this increase was not significant. Generally, fatty acid production is associated with the composition of grapes and fermentation conditions, and the aroma is usually unpleasant, ranging from sweaty and cheesy to goaty and rancid [7]. Acids are generated through the hydrolysis of acyl-S-CoA, either via the synthetic anabolic pathway, which is responsible for the formation of most fatty acids, or through the β-oxidation of lipids [28]. Furthermore, it has been proven that as the length of its chain increases, the volatility decreases and the smell changes from acidic to rancid [29]. The results obtained for both wines (control and EM) showed a majority of short- and medium-chain acids. However, it is worth highlighting the presence of isobutyric acid only in EM wine, since isobutyric is a short-chain fatty acid that can contribute to enhance the aroma profile of wines. This acid is known for its pungent, sweaty or rancid aroma, and in high concentrations, it can contribute to off-flavours in wines. However, in small amounts (as is the case with EM wine), it may add complexity to the aroma profile, enhancing the overall sensory experience.

Ketones are organic compounds that incorporate a carbonyl functional group. They are produced in relatively small amounts in grapes, and usually do not play a key role in the creation in of varietal aroma in wines. Usually, not many ketones are found in wines but those present in grapes usually survive to fermentation. Three ketones were found in Monastrell wines, two in EM wines (2-butanone and 2,3 butanedione) and one in control wine (2-octanone) (Table 1). The 2-butanone identified in wines from extended macerations is associated with a sweet and fruity aroma, contributing to the overall aromatic profile of the wine. Additionally, 2,3-butanedione (also known as diacetyl), identified only in the EM wine, is one of the most important compounds responsible for the characteristic aromas and flavours associated with yeasty, nutty, toasty and buttery notes in wines [30]. Diacetyl is formed as an intermediate metabolite in the reductive decarboxylation of pyruvic acid to 2,3-butanediol [31]. Diacetyl was only found in EM wines probably due to an excess of pyruvate [30]. Therefore, EM might be a good oenological technique to increase the variability and concentration of ketones in the resulting wines, modifying the aromatic profile of the wines thanks to greater aromatic complexity.

Furfural and dimethyl sulphide were the only two volatile sulphur compounds identified in Monastrell wines, at slightly higher concentrations than in control wine (Table 1). Dimethyl sulphide (DMS) is a light sulphur compound that has been described in a wide range of wine studies [32–34]. The presence of DMS in red wines is often associated with undesirable odour notes of asparagus, cauliflower, canned corn and quince [35]. It is known that DMS content in young wines is typically low and usually with negligible sensory relevance. Conversely, aged wines are characterized by much higher levels of DMS [36]. Although our results showed a higher concentration in control wine (11.2 µg L$^{-1}$) compared to EM wine (8.4 µg L$^{-1}$), these values were much lower than the olfactory threshold of DMS established at 27 µg L$^{-1}$ in red wines [37], but this seems to be strongly dependent on the matrix wine, reaching an estimate of up to 60 µg L$^{-1}$ [38]. Another study reported a positive contribution to wine aroma, enhancing the fruity flavour when DMS concentrations were lower than the odour threshold. Therefore, our findings suggest that employing the extended maceration technique might be applied with the aim of reducing the concentration of DMS in wines suitable for extended aging.

One of the most important groups of volatile compounds in wines are terpenes. They are varietal compounds obtained from grapes through direct extraction, as well as from the breakdown of glycoconjugates in the skin and in the solid parts of berry cells during the process of maceration [25]. These compounds actively participate in the flavour profile of wines and are responsible for floral, rose-like, citrusy, lemongrass and spicy aromas, and their importance is related to their low odour threshold [39]. Additionally, terpenes are key odorants in aromatic white grape varieties of Vitis vinifera, such as Muscat of Alexandria, Riesling and Gewürztraminer [40]. Although the Monastrell red variety is not commonly known as terpenic, limonene and eugenol were identified only in the control wines. Both terpenes have been identified previously in Monastrell wines and should

be common in young wines [15,19]. Limonene has been described previously by terms such as lychee, citrus, rose, sweet or floral [25]. Previous studies showed that the presence of the skins during the maceration process could decrease the levels of terpenes in the resulting wines [26,41]. These results are in agreement with those observed in this study, where the concentration of both terpenes (limonene and eugenol) disappeared after the EM. However, other studies showed that an extended skin contact period caused an increase in terpene concentration [25]. Thus, the final concentration of terpenes might be affected by the grape variety after EM. Lastly, a butyrolactone was identified, which is a compound not previously classified within the identified families (Table 1). Lactones, generally recognized for their fruity aroma, are likely to contribute significantly to the overall wine aroma. Butyrolactone, commonly detected among volatile compounds in red wines, is notably associated with a buttery scent. In contrast to our results, refs. [2,7] reported an increase of butyrolactone concentration after 15 days of EM in Karaoğlan and 30 days of EM in Vranec wines, respectively.

The total alcohol concentration decreases significantly after 146 days of EM (Figure 1). It is known that the extraction of aroma compounds from grapes into the must is mainly a diffusion process [2,41,42]. In this way, several studies have reported a more efficient extraction of grape aroma with longer maceration time. More concisely, ref. [2] observed an increase in the total content of alcohols in Vranec wines as maceration progressed, up to one month, and then decreased, remaining the same control concentration. This increase in the total alcohols content during the first stages of EM has also been observed in white varieties such as Chenin Blanc [25]. In this regard, our findings are consistent with those presented by [14] in EM wines from Verdeca and Minutolo grapes. These authors described an increase in total alcohol content during the early stages of maceration in both wines, and after a few weeks it decreased (Verdeca) or remained (Minutolo) at a concentration similar to control wines. Therefore, the trend showed an increase in the content of volatile alcohol compounds during the first stages of maceration due to a diffusion effect from the grape skin to the must/wine. Subsequently, long-extended maceration reduces the concentration of alcohols as they can be sorbed on macromolecules [43] or oxidized into aldehydes [14]. Furthermore, it is known the EM may also inhibit the Ehrlich pathway, a process responsible for the production of higher alcohols from yeast-mediated amino acid catabolism, which is a primary route for the genesis of these compounds [7].

Acids concentration was significantly lower in EM wines (119 $\mu g \cdot L^{-1}$) compared to control wines (194 $\mu g \cdot L^{-1}$). Similar results were reported by [14]; these authors observed a marked decrease in acid concentration from the first month of maceration, which remained stable throughout EM. In Monastrell wines, only terpenes were identified, and in a very low concentration (52 $\mu g \cdot L^{-1}$) in the control wine, with no terpenes detected in the EM after 6 months. This low concentration of terpenes in control wine are similar to those observed by [14] in Verdeca wines (30 $\mu g \cdot L^{-1}$), which remained almost unchanged throughout maceration. Furthermore, in EM wines, these authors also reported lower amounts of terpenes, sometimes even reaching zero. Similarly, ref. [24] found a reduction in norisoprenoids in prolonged macerated wines. Sulphur compounds concentration showed a significant decrease in EM wines. Their presence indicates reductive processes, suggesting that monitoring oxygen supply and redox equilibria throughout the EM may not be necessary, as these compounds decrease in EM wine. Contrary to what was observed with the previously mentioned groups of volatile compounds (alcohols, acids, terpenes and sulphur compounds), control and EM wines did not show significant differences in the total concentrations of esters and aldehydes and ketones after 146 days of extended maceration (Figure 1). Although the differences between treatments were not statistically significant, the total concentration of esters was higher in the control wine (7268 $\mu g \cdot L^{-1}$) than in the EM wine (6429 $\mu g \cdot L^{-1}$). Aldehydes and ketones were also higher in the control wine (22.8 $\mu g \cdot L^{-1}$) compared to the EM wine (14.8 $\mu g \cdot L^{-1}$). According to the previous literature and in line with our results, maceration length does not directly affect the content

of esters in wine [2,44], and could even generate a decrease in acetates of higher alcohols and ethyl esters of fatty acids as were observed previously by [2].

### 3.2. Sensory Evaluation of Monastrell Wines

After twelve month of bottling, Monastrell wines were subjected to sensory discrimination testing using a descriptive and a triangle test by a consumer panel. These sensory analyses were conducted on the same day as the volatile compound analyses SPME/GC–MS. The results obtained for the triangle test for Monastrell wines are presented in Table 2. Ten out of the eighteen consumers (55%) were able to differentiate the control from the EM wines. Therefore, in a blind triangular discriminative test, half of the taster panel were able to distinguish the wines. Thus, the results indicated that wines resulting from both winemaking processes have a sensorial profile with similar general characteristics. Moreover, twelve tasters (67%) chose the EM wine as their preferred wine.

**Table 2.** Sensory analysis (triangular discriminative and preference test) for 'Monastrell' extended-maceration wines after twelve-month bottling.

| Year | Triangular Test | Identifications | Preferences |
|------|-----------------|-----------------|-------------|
| 2024 | Control vs. EM | 10/18 * | EM (12) |

* Significant differences at $p \leq 0.05$.

Descriptive analysis is depicted in Figure 2. Control wines exhibited higher scores in most of the evaluated parameters (excepted vegetal, spice aromas and mouthfeel intensity), although only the parameters of colour intensity, tonality and aroma intensity were statistically significant. The panel of tasters perceived a higher colour intensity and hue in the control wine. This higher hue value indicates that, although the control wine had a deeper colour, it was also browner with a lower intensity of red coloration. Similarly, ref. [45] reported a significant decrease in colour intensity in six-month EM Zifandel wines. In contrast, these authors did not report any increase in the wine's hue. The decrease in colour intensity with increasing maceration time has also been documented in other grape varieties, including Cabernet Sauvignon and Malbec [45]. However, this reduction in colour intensity was not observed by [6] after 90 days of EM in Aglianico di Taurasi, nor by [46] after 64 days of EM in Merlot. So, the effect of post-fermentative EM on the chromatic composition of red wines depends on the variety used. It is worth noting the lower perception of astringency that tasters have reported in the EM wine compared to the control wine. Generally, published studies agree that EM should increase astringency in the resulting wines [5,9,10,14,47]. However, our findings revealed divergent results, as tasters perceived the EM treatment to be less astringent than the control wine. Consistent with our findings, other studies have also reported a reduction in astringency in wines from different red grape varieties. For instance, ref. [48] observed reduced astringency in Syrah wines after 30 days of EM, while [6] noted similar effects in Aglianico di Taurasi wines after 90 days of EM. The reduced perception of astringency evaluated by tasters in EM wines could be explained by the increase in the (+)-catechin/(−)-epicatechin ratio in wines with longer maceration (due to the slow extraction of catechins during this process) [6]. Knowing that epicatechin is more astringent than its chiral isomer [49], this would contribute to a greater sensation of roundness in the wine. On the other hand, EM might have favoured higher extraction of polysaccharides from grapes, reducing the astringency and enhancing the body and sweetness sensation [50].

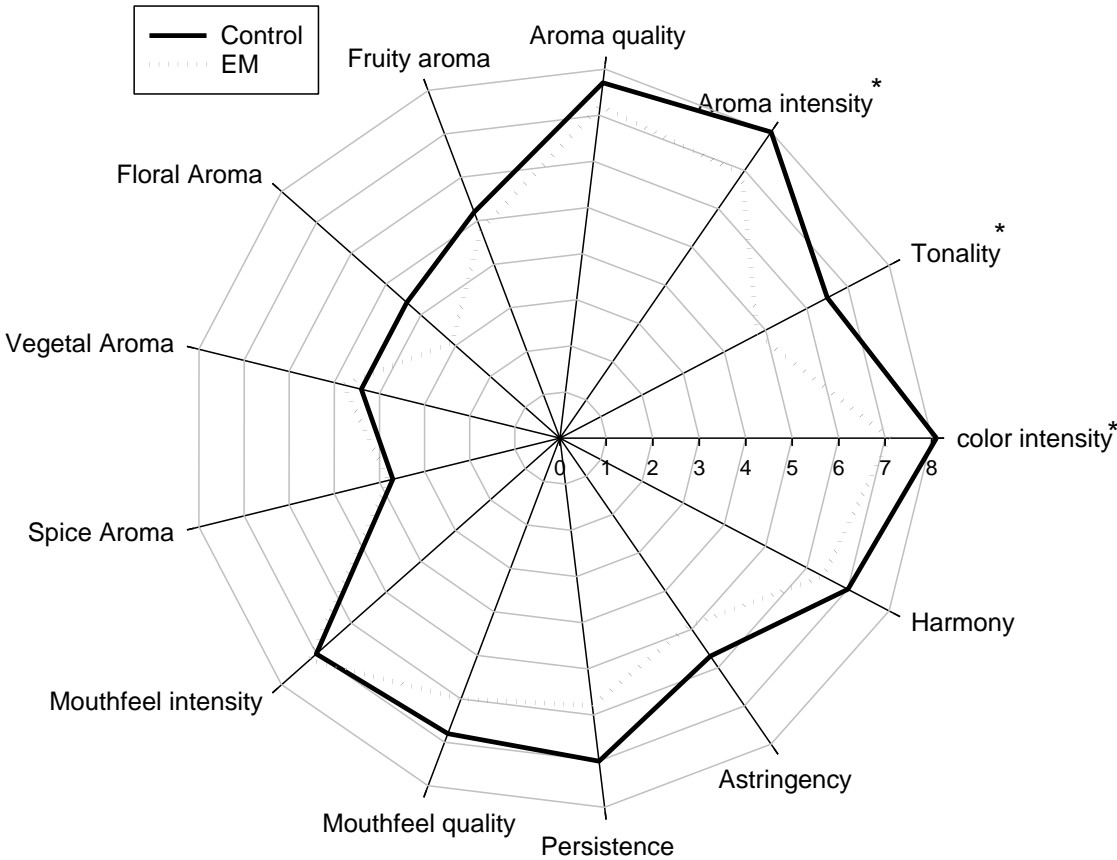

**Figure 2.** Sensory analysis of extended-maceration Monastrell wines assessed by a consumer panel after twelve months of bottling. * Significant differences at $p \leq 0.05$.

## 4. Conclusions

In this work, we explored the prolonged maceration of Monastrell wine since it has not been described any study so far. A total of 43 individual volatile compounds were detected in Monastrell wines elaborated with different maceration times, 11 days for the control wine and 146 days for EM. Monastrell wines presented a complex aroma profile across various chemical groups of aroma compounds: alcohols, esters, acids, aldehydes and ketones, terpenes, sulphur compounds and others. The results suggest that after 146 days of contact with skin and seeds maceration significantly impacted the volatile and sensory profiles of Monastrell wines, significantly reducing the concentrations of alcohols, acids, terpenes, sulphur compounds and others. However, not all individual compounds showed this trend; certain volatiles such as phenylethyl alcohol, 1-ethyl-1-hexanol and ethyl decanoate increased in extended-maceration wines. According to triangular sensory analysis, wines presented different profiles as identified by the tasters (10 out of 18), although it is worth noting that more tasters chose the EM wine as their preferred option (12/18). EM wines presented less colour intensity, hue and astringency (which is different than expected since extended maceration is related to astringency). These findings underscore the potential for optimizing maceration duration to enhance wine profiles without compromising quality. Future research aims to refine these techniques for broader application in the wine industry, emphasizing improvements in aroma, colour stability and sensory attributes of Monastrell wines.

**Author Contributions:** A.M.-M.: conceptualization, investigation, formal analysis, methodology, writing—reviewing and editing original draft. R.T.-G.: conceptualization, investigation, sample preparation, formal analysis, methodology, writing—original draft. A.B.B.-O.: conceptualization, investigation, formal analysis, methodology, writing—review and editing. E.G.-P.: conceptualization, investigation, methodology, writing—review and editing. J.E.Y.: conceptualization, investigation, formal analysis, methodology, writing—review. F.V.: conceptualization, investigation, formal analysis, methodology, writing—original draft, writing—reviewing and editing the final manuscript. All authors have read and agreed to the published version of the manuscript.

**Funding:** This research received no external funding.

**Institutional Review Board Statement:** Not applicable.

**Informed Consent Statement:** Not applicable.

**Data Availability Statement:** The data are contained within the article.

**Acknowledgments:** We want to thank Jose Ramon Rabadán Plaza for his technical support.

**Conflicts of Interest:** The authors declare no conflicts of interest.

**Appendix A**

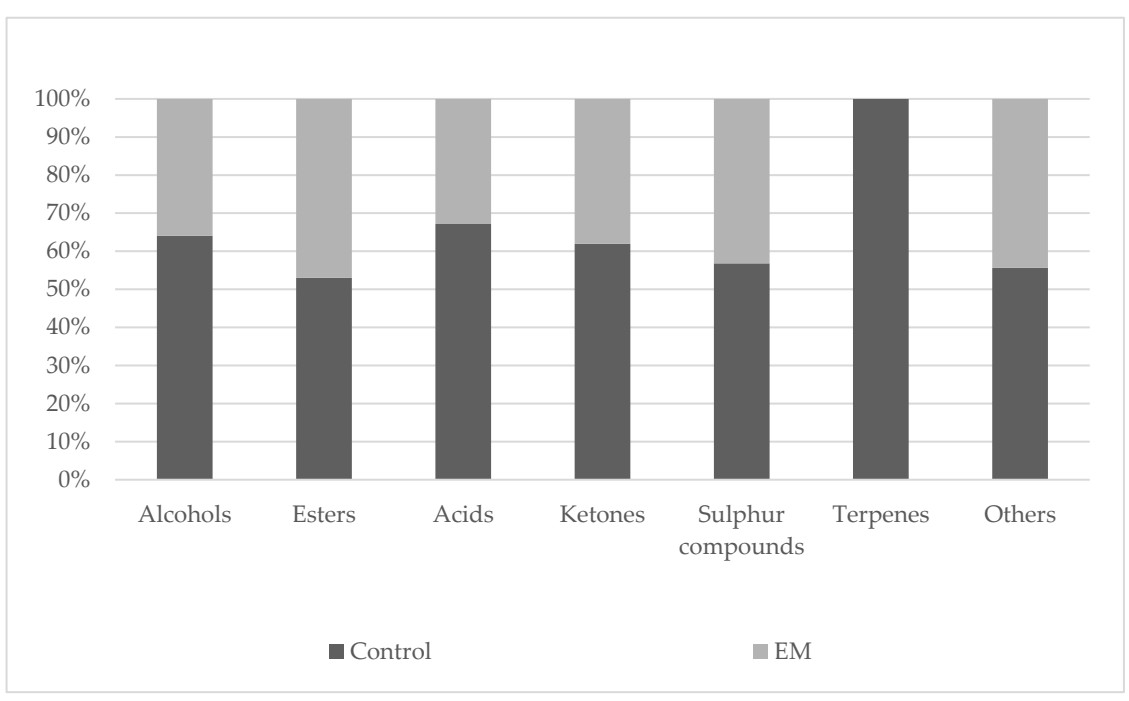

**Figure A1.** Distribution of percentages of chemical families identified between control and extended-maceration (EM) wines by solid-phase microextraction SPME/GC-MS.

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
