# Peer review of "Exploring the Impact of Extended Maceration on the Volatile Compounds and Sensory Profile of Monastrell Red Wine"

_fermentation, doi:10.3390/fermentation10070343_

Round 1
Reviewer 1 Report
Comments and Suggestions for Authors
The manuscript discusses how extended maceration affects the aromatic qualities and sensory characteristics of Monastrell. It's well-written and comprehensible, but there are some questions the authors need to address to fulfill the publication standards.
· What motivated the choice to extend maceration for 146 days instead of the more typical industry practice of 30 or 60 days?
· Control wines should not be capitalized. Modify throughout the manuscript.
· Grape varieties should be presented without quotation marks. Modify throughout the manuscript.
· Line 81-83. Sentence lacks coherence. Please include the matrix the study was conducted on for comparison.
· Line 92-94. Sentence is unclear. Rewrite.
· Authors should include initial chemical composition of the grapes (Brix, pH, TA, Malic, YAN…) and final composition of the wines (%Alc, pH, TA, Malic…)
· Were any adjustments done to the must regarding pH, nitrogen…?
· Was MLF performed? Information should be added to the manuscript whether or not MLF was performed.
· What was DO at bottling? Bottle type and closure should be included.
· Line 119- Analysis done on triplicate… Does this refer to fermentation replicates or bottle replicates of each fermentation?
· Line 176-177- Authors should use the same format as lines 172-173 for easier comparison.
· Line 188-189- Authors state triplicate test but M&M statistical analysis claims quintuplicate. Clarify.
· Line 189- Which statistical analysis was carried out to determine significance? Information should be included.
· Line 214-219. Sentence is unclear. Rephrase.
· Line 237- Increased
· Line 258- Hydrolyzes
· Line 310- Terpenes should not be capitalized.
· Line 327-328. Sentence lacks coherence. Clarify
· Line 351- Missing reference
· How was preference evaluated during a triangle test? Drawing conclusion on preference with only 18 panelists is not accurate.
· Line 392-397. This paragraph is unclear. Compares control results with previous EM research results. Rewrite and clarify.
· Did authors perform any color analysis to compare to panelist perception?
· A multi-factor analysis should be conducted to correlate chemistry data with sensory data, ensuring that the results from both sections align.
Comments on the Quality of English LanguageMinor editing of English language required. Few typos throughout the manuscript
Reviewer 2 Report
Comments and Suggestions for Authors
Dear Authors,
I am writing to review the manuscript entitled “Exploring the Impact of Extended Maceration on the Volatile Compounds and Sensory Profile of ‘Monastrell’ Red Wine”.
This manuscript contains the crucial role of volatile organic compounds in defining the aromatic profile of wines. The primary objective of the study is to explore the impact of an extended maceration treatment of 146 days on the aromatic and sensory properties of selected ‘Monastrell’ red wine.
The authors identified 43 aroma compounds across various chemical classes using SPME/GC-MS. The study found that maceration treatment generally decreased both the number and total relative concentration of volatile organic compounds. The authors find that the concentration of alcohols, terpenes, and sulfur compounds decreased in extended maceration-treated wines compared to control wines, while certain compounds like 2-ethyl-1-hexanol, phenylethyl, and ethyl decanoate increased with prolonged maceration.
From a sensory perspective, the triangular test indicated a significant preference for extended maceration wines, with identification in 10 out of 18 trials. However, descriptive analysis revealed that extended maceration wines scored lower in appearance, aroma, and taste compared to control wines. These findings suggest that while extended maceration can alter the volatile organic compounds profile and sensory characteristics of ‘Monastrell’ red wine, there is a need to optimize maceration time to enhance volatile organic compounds content without compromising sensory quality.
I have thoroughly reviewed the manuscript and believe it provides valuable insights into the effects of extended maceration on wine quality. The study is well-conducted and contributes significantly to the field of enology. I recommend the manuscript for publication following minor revisions to address the questions and suggestions outlined in this cover letter.
Abstract:
Lines 13-15: Could you rewrite this sentence in something like this: “The primary aim of the study was to investigate whether a 146-day extended maceration (EM) treatment positively affects the aromatic and sensory properties of ‘Monastrell’ red wine.”
Line 15: aroma compounds? Maybe aromatic compounds?
Line 19: Could you rewrite “Control wines” as “control wines”
Line 23: Perhaps you should begin with “Moreover” instead of “On the other hand” because the previous sentence started with “Conversely,” which has the same meaning as “On the other hand.”
Line 24: Maybe you should delete “lines”
Keywords:
Maybe you should add the name of the used sort “Monastrell” as the last keyword
Introduction:
Line 37: C13 – you should write 13 in subscript as C13
Line 56: CO2 – you should write 2 in subscript as CO2
Line 76: You should delete (2014) from this sentence it is usual to write “Petropulos et al. [2]”; the same comment for the lines 81, 115, 150 etc.
Line 100: SO2 – you should write 2 in subscript as SO2
Why you use point at the end of the title of sub-paragraph? For example: “2.3. Sensory analysis.” (Line 144), I recommend to you write all sub-paragraph titles without point
Line 159: Please check this “p ≤ 0.05 level.3.”
Lines 234-235; 351, 354: Please carefully check all units, because you must correct it as: µg L-1
Conclusion:
I recommend to you carefully check all this segment and you should not use references in your conclusion based on your results… Please made clear conclusion based on the influence of extended maceration treatment on tested wine varieties. And future perspectives and the most important importance of your study for implementation in wine industry.
Appendix B:
Figure A1. you should provide figure with higher resolution and also you should use the same font (Palatino linotype) which is mandatory for this Journal.
References:
Additionally, please thoroughly check all references in the reference list and ensure that all Latin names are correctly written in italics.
For example reference number 1 should be cited as:
1. Martínez-Moreno, A.; Pérez-Á lvarez, E.P.; López-Urrea, R.; Paladines-Quezada, D.F.; Moreno-Olivares, J.D.; Intrigliolo, 473 D.S.; Gil-Muñoz, R. Effects of deficit irrigation with saline water on wine color and polyphenolic composition of Vitis vinifera L. cv. Monastrell. Sci. Hort. 2021, 283, 110085.
All volumes should be written in italic.
Line 520: You should use only coma “,” after volume number not “:”.
